# Influence of Different Drying Methods on Anthocyanins Composition and Antioxidant Activities of Mangosteen (*Garcinia mangostana* L.) Pericarps and LC-MS Analysis of the Active Extract

**DOI:** 10.3390/foods12122351

**Published:** 2023-06-12

**Authors:** Nur Izzati Mohamed Nawawi, Giroon Ijod, Faridah Abas, Nurul Shazini Ramli, Noranizan Mohd Adzahan, Ezzat Mohamad Azman

**Affiliations:** 1Department of Food Technology, Faculty of Food Science and Technology, Universiti Putra Malaysia, Serdang 43400, Selangor, Malaysia; nurizzatinawawi@gmail.com (N.I.M.N.); giroons25292@gmail.com (G.I.); noraadzahan@upm.edu.my (N.M.A.); 2Department of Food Science, Faculty of Food Science and Technology, Universiti Putra Malaysia, Serdang 43400, Selangor, Malaysia; faridah_abas@upm.edu.my (F.A.); shazini@upm.edu.my (N.S.R.)

**Keywords:** mangosteen pericarps, freeze-drying, oven-drying, sun-drying, anthocyanins, natural colorants

## Abstract

Mangosteen pericarps (MP) often end up as agricultural waste despite being rich in powerful natural antioxidants such as anthocyanins and xanthones. This study compared the effect of different drying processes and times on phenolic compounds and antioxidant activities of MP. Fresh MP were subjected to 36 and 48 h of freeze-drying (−44 ± 1 °C) and oven-drying (45 ± 1 °C), and 30 and 40 h of sun-drying (31 ± 3 °C). The samples were analyzed for anthocyanins composition, total phenolic content (TPC), total flavonoid content (TFC), antioxidant activities, and color characteristics. Analysis of liquid chromatography-mass spectrometry (LC-MS) with electrospray ionization identified two anthocyanins in MP: cyanidin-3-*O*-sophoroside and cyanidin-3-*O*-glucoside. Overall, the drying process, time, and their interactions significantly (*p* < 0.05) influenced the phenolic compounds, antioxidant activities, and color in MP extracts. Both freeze-drying after 36 h (FD36) and 48 h (FD48) possessed significantly (*p* < 0.05) higher total anthocyanins (2.1–2.2 mg/g) than other samples. However, FD36 was associated with significantly (*p* < 0.05) higher TPC (~94.05 mg GAE/g), TFC (~621.00 mg CE/g), and reducing power (~1154.50 μmol TE/g) compared to FD48. Moreover, FD36 is more efficient for industrial applications due to less time and energy consumption. Subsequently, obtained dried MP extracts could be further utilized as an alternative to synthetic food colorants.

## 1. Introduction

Mangosteen (*Garcinia mangostana* L.) is an endemic evergreen tree species grown in tropical rainforest countries, such as Malaysia, Thailand, and Indonesia [1,2]. Referred to as the “Queen of Fruits”, it generally has an exocarp of dark-purple to reddish color, with juicy white fleshy pulp enclosed within. Mangosteens can be consumed as fresh or processed into commercial products for various purposes such as dietary supplements, pharmaceuticals, and cosmetics products [3].

For the past hundred years, mangosteen has been used as traditional medicine by Southeast Asians and South Americans for the treatment of skin infections, external wounds, suppurations, diarrhea, and chronic ulcers [4]. Many studies reported that mangosteen contains various health benefits such as strong antioxidant, anti-microbial, anti-inflammatory [5], anti-diabetic [3], and anti-cancer effects, regulation of menstrual problems and blood pressure, weight loss, as well as boosting immunity [6]. However, compared to the pulp, the mangosteen pericarp (MP) has 20 times higher antioxidant activity and 10 times more phenolic compounds [7].

In Malaysia, around 70% of mangosteens are pericarps and often end up as agricultural waste despite being rich in powerful natural antioxidants such as anthocyanins and xanthones [8,9]. Utilizing this type of resource as an alternative to food colorants is aligned with circular economy concepts as summarized by Zainal Ariffin et al. [10]. In addition, studies are currently being conducted to seek out natural, safer, and environmentally friendly pigments, as synthetic dyes have been associated with negative effects on human health [11,12]. The dark-purple or reddish color of MP is contributed by polyphenolic pigments, namely anthocyanins [13]. Based on availability and anthocyanin concentration, MP are a promising source of natural colorants.

Anthocyanin pigments are important to food quality because of their contribution to color and appearance. Major anthocyanins found in the MP are cyanidin-3-*O*-sophoroside (C3S; 76.1%) and cyanidin-3-*O*-glucoside (C3G; 13.4%) [14]. Pelargonidin-3-*O*-glucoside (P3G; 6.2%) was found at a lower concentration by Zarena and Udaya Sankar [15]. However, the issues that arise in the utilization of MP wastes are the natural antioxidants from the active compound, which tend to be lower in practicality, color stability, solubility, and bioavailability [16]. Thus, considering that the rich antioxidant compounds in MP are easily degraded at high temperature, suitable drying conditions prior to extraction should be considered for application in food systems [17].

Drying is one of the traditional methods of preserving seasonal fruit and vegetables while maintaining or enhancing their overall quality. This process reduces the moisture up to a safe limit between 6% to 11% (*w*/*w*) for better stability of phytochemicals while slowing down enzyme activity and microbial growth [18,19]. Since phytochemical degradation occurs more quickly in environments with high-water activity, it is more effective to extract bioactive components from the dried mass as opposed to fresh material [20].

Sun-drying has been practiced around the world for centuries. It utilizes solar energy to lower the moisture content of crop products by providing heat to evaporate moisture, while the available wind helps in removing moisture and speeding up the drying process [21]. After all, this drying method has various drawbacks including unpredictable drying times, high labor costs, requirements for large spaces, and contamination by insects and foreign bodies [22]. Oven-drying is the most widely used drying technique in the food industry because it is inexpensive and easily applied while extending the shelf-life of the product. This method typically operates from ambient temperature to 300 °C, where thermal energy is transferred to the chamber load through convection and radiation [23]. Freeze-drying removes water through sublimation directly from the ice phase into the gas phase under low pressure [24]. It offers great preservation of natural color by preventing oxidation damage and retaining volatile compounds while minimizing chemical compound changes. Despite its high operating costs, freeze-drying is effective in retaining heat-sensitive compounds and high-quality products [25].

Since mangosteen is available seasonally, drying the pericarps is an alternative to prolong the shelf life and enable the powders to be utilized as food colorants or additives in other food matrices [20,26]. However, it should be highlighted that drying conditions will affect the quality characteristics of dried fruits. Phytochemicals such as anthocyanins and phenolic compounds are highly susceptible to degradation, thus reducing their antioxidant activity, particularly their functional characteristics [27]. Focusing on that, this is the first study aiming to investigate the effect of drying methods on the anthocyanin compositions, antioxidant activities, total phenolics, total flavonoids, and color characteristics of MP. In addition, Pearson correlation and principal component analysis (PCA) were carried out to further understand the correlations between the tested parameters.

## 2. Material and Methods

### 2.1. Chemicals

Trolox, 2,2-diphenyl-1-picrylhydrazyl (DPPH), and iron (III) chloride hexahydrate (FeCl_3_·6H_2_O) were purchased from Sigma-Aldrich (UK); Folin-Ciocalteu reagent, 2,4,6-tripyridyl-s-triazine (TPTZ) and gallic acid (98%) were purchased from Merck (Darmstadt, Germany). Standards of C3S (96%), C3G (96%), and P3G (96%) were purchased from ExtraSynthese Ltd. (Genay, France). Other chemicals and solvents used were of analytical grade and bought from Fisher Scientific (Leicestershire, UK).

### 2.2. Sample Preparation

Fresh mangosteens (*Garcinia mangostana* L.) were purchased from a local farm in Hulu Langat, Selangor, Malaysia. The selection of fresh fruits was based on maturity levels ranging from 1 to 6, where maturity index 6 (purple-black) was selected based on visualization characteristics. The mangosteens were cleaned and washed from yellow saps and dirt. MP were cut into four to separate the flesh from the pericarps prior to the drying process.

*Freeze-drying*: 600 g of fresh MP was freeze-dried in a freeze-dryer (Labconco, Kansas City, MO, USA) at 0.13 mbar pressure and −44 ± 1 °C inside the vacuum chamber for 36 and 48 h, constituted as FD36 and FD48, respectively.

*Oven-drying*: 600 g of fresh MP was spread on the tray and placed in a conventional laboratory oven (Oven Binder ED23, Tuttlingen, Germany) and dried at 45 ± 1 °C for 36 and 48 h, constituted as OD36 and OD48, respectively.

*Sun-drying*: 600 g of fresh MP was spread over the tray and placed under direct sunlight from 8 am to 6 pm at an average temperature (31 ± 3 °C). MPs were turned every one-hour interval for uniform drying for 3 days (30 h) and 4 days (40 h), constituted as SD30 and SD40, respectively.

For the preparation of dried MP powder, dried MP were ground for 30 s and sieved to pass through a 0.841 mm sieve (20 mesh). All samples were stored in a plastic container and kept at −20 °C for further analysis.

### 2.3. Determination of Moisture Content and Water Activity

*Moisture content*: The moisture content of dried MP was determined using a moisture analyzer (MX-50, Tokyo, Japan). Measurements were performed in duplicates.

*Water activity*: The MP’s water activity (A_w_) was determined using a water activity meter (Aqualab, Pullman, WA, USA). Approximately, 1 g of dried MP was placed in a plastic container and inserted in the instrument chamber. A_w_ of each sample was recorded in duplicates.

### 2.4. Preparation of Dried Mangosteen Pericarps (MP) Extracts

Dried MP extracts were obtained according to the method previously described by Azman et al. [28] with slight modifications, whereby different ratios of dried MP and solvent were used to favor better anthocyanins extraction. Approximately, 10 g of dried MP was mixed with 100 mL of 50% ethanol. The mixture was shaken at 50 °C for 2 h in a water bath (WITEG Labortechnik, Wertheim, Germany) at 180 rpm. The solution was vacuum- filtered using a Buchner funnel and Whatman No. 1 filter paper (Whatman, Buckinghamshire, UK). The extracts were centrifuged at 2500 rpm for 15 min to separate the clear extract and precipitate. Samples were then kept at −20 °C for further analysis.

### 2.5. Determination of Total Monomeric Anthocyanins Content (TMAC)

In preliminary studies, total monomeric anthocyanins content (TMAC) was assessed using the pH differential method [29] with some modifications, where different dilution factors and volumes of extract were used. The samples were prepared following the procedure outlined by Azman et al. [28] with minor modifications. Specifically, 0.6 mL of 10-fold diluted extracts was mixed with 1.2 mL of corresponding buffer (pH 1.0 and pH 4.5) and allowed to equilibrate in the dark for 30 min. TMAC was then calculated as mg cyanidin-3-*O*-glucoside equivalents per gram as follows:(1)$$C_{anthocyanins}=\frac{A\times MW\times DF\times 10^{3}}{\varepsilon \times 1}$$
where *C_anthocyanin_* = total monomeric anthocyanins concentration (cyanidin-3-*O*-glucoside equivalents, mg/L); *A* (absorbance) = (*A*_520nm_ − *A*_700nm_) pH 1.0 − (*A*_520nm_ − *A*_700nm_) pH 4.5; *MW* (molecular weight) = 449.2 g/mol for cyanidin-3-*O*-glucoside; *DF* = dilution factor; 1 = path length in cm; *ε* = 26,900 molar extinction coefficient in L/mol/cm for cyanidin-3-*O*-glucoside; and 10^3^ = factor for conversion from g to mg (Lee et al., 2005) [29]. The anthocyanins content was then calculated by Equation (2) as follows:(2)$$Anthocyanin\;content\left(\frac{mg}{g}\right)=\frac{\left(\frac{mg}{L}\right)\times extract\left(L\right)}{sample\left(g\right)}$$

The absorbance of the samples at 520 and 700 nm were determined using a spectrophotometer (Thermo Fisher Scientific, Waltham, MA, USA).

### 2.6. Identification of Phenolic Compounds by Liquid Chromatography-Mass Spectrometry (LC-MS)

A Dionex Ultimate 3000 Rapid Separation (RS) UPLC system (Thermo Fisher Scientific Inc., USA) with a Thermo Scientific Q Exactive Orbitrap Hybrid Tandem mass spectrometer and Heated-Electrospray Ionisation (H-ESI II) were used to identify anthocyanins profiles in dried MP using the method by Azman et al. [19] with slight modifications. Chromatography was carried out on a Purospher START RP18 end-capped column (5.0 µm, 4.6 mm i.d × 250 mm) temperature maintained at 30 °C. The mobile phases consisted of Solvent A (formic acid:water; 2:98, *v*/*v*) and Solvent B (methanol; 100, *v*/*v*) at a flow rate of 0.8 mL/min with 10 µL injection volume. The gradient elution program was used as followed: 0–19 min, 15% B; 19–38 min, 35% B; 38–50 min, 60% B; 50–56 min, 80% B; 56 min, 15% B. Mass spectrometry (MS) spectra were operated in the positive and negative ion mode between *m*/*z* 100 and 1500 at a scan resolution of 70,000 (full MS scan) and 35,000 (ddMS2 scan). The Qual Browser of the Xcalibur software (Thermo Scientific, USA) was used to analyze the acquired data for MS analysis.

### 2.7. Quantification of Anthocyanins by High-Performance Liquid Chromatography (HPLC) 

HPLC analysis of anthocyanins was based on a method by Azman et al. [30]. Briefly, the mobile phase consisted of 2% (*v*/*v*) formic acid (Solvent A) and 100% (*v*/*v*) methanol (Solvent B). The gradient condition was 15% (B) at 0 min, 35% (B) at 15 min, 60% (B) at 30 min, and end at 80% (B) at 40 min. A Purospher STAR RP18 end-capped column (250 mm × 4.6 mm i.d., particle size of 5 µm, Merck, Darmstadt, Germany) in a Perkin Elmer Series 200 HPLC system (PerkinElmer Inc., Bridgeport Avenue, Shelton, CT, USA), equipped with a Perkin Elmer Series 200 UV/Vis detector, Perkin Elmer Series 200 pump, Shimadzu CTO-10A column oven with a manual injector, and Perkin Elmer Series 200 vacuum degasser. The column temperature and flow rate were set up at 30 °C and 1.0 mL/min, respectively. Notably, ethanol was removed before injection into the HPLC system. The injection volume was fixed at 20 µL and the period of analysis was 45 min. A wavelength of 520 nm was used to detect the anthocyanins.

The calibration curve for each anthocyanin was plotted against the concentrations range of 0.1–0.02 mg/mL. Table 1 shows the determination coefficient (R^2^), the limit of detection (LOD), and limit of quantification (LOQ), which were calculated as:(3)$$LOD=\frac{{3S}_{a}}{b}\;and\;LOQ=\frac{{10S}_{a}}{b}$$
where $S_{a}$ is the standard deviation of the response and *b* is the slope of the calibration curve [31].

### 2.8. Determination of Total Phenolic Content (TPC)

Total phenolic content (TPC) was determined by the Folin–Ciocalteu method [28]. Firstly, 20 µL of the 10-fold diluted extract was mixed with 1.58 mL of distilled water and 100 µL Folin–Ciocalteau reagent and vortexed for 10 s. After 8 min, 300 µL of the 7.5% (*w*/*v*) sodium carbonate solution was added and the mixture was vortexed for 10 s and left in the dark for 2 h. 

Gallic acid was dissolved in water and sonicated in an ultrasonic bath, then used as a stock standard solution for the calibration curve (1.0–0 mg/mL). The absorbance was measured at 765 nm using a spectrophotometer, and the results were expressed as milligrams of gallic acid equivalents per gram of the samples (mg GAE/g). Duplicate measurements were taken, and mean values were calculated.

### 2.9. Determination of Total Flavonoid Content (TFC)

The total flavonoid content (TFC) was determined using the method by Yea et al. [32]. An amount of 1 mL of 100-fold diluted extract was mixed with 4.0 mL distilled water. Then, 300 µL of 5% (*w*/*v*) NaNO_2_ and 300 µL of 10% (*w*/*v*) AlCl_3_ were added, and the solution was to stand for 5 min. After this, 2.0 mL of 1 M NaOH was added. The catechin solution (300–0 mg/L) was used to plot a standard curve, and the absorbance was measured at 510 nm using a spectrophotometer. The results were expressed as milligrams of catechin equivalent per gram of the samples (mg CE/g).

### 2.10. Determination of Trolox Equivalent Antioxidant Capacity (TEAC)

The antioxidant capacity of MP extracts was analyzed according to the method of Senevirathna et al. [33] with a slight modification, where different dilution factors and volumes of extract were used. Briefly, a mixture was prepared by combining 0.05 mL of diluted extracts (50-fold) or Trolox solution with 1.95 mL of a 0.15 mM DPPH solution in methanol. The concentration range of Trolox used was 2000–0 µM. Then, the mixture was vortexed for 10 s and incubated for 30 min in the dark. The absorbance was measured using a spectrophotometer at 517 nm. The results were expressed as µmol of Trolox equivalents per gram of the samples (µmol TE/g). Duplicate measurements were taken, and mean values were calculated.

### 2.11. Determination of Ferric Reducing Antioxidant Power (FRAP)

The ferric reducing antioxidant power (FRAP) was analyzed according to Senevirathna et al. [33] with some modifications, whereby the incubation time was increased to 30 min and the extracts were diluted prior to mixing with FRAP reagent. The FRAP reagent was prepared by mixing 300 mM/L acetate buffer (pH 3.6), 10 mM/L of TPTZ, and 20 mM/L FeCl_3_·6H_2_O with a ratio of 10:1:1 at 37 °C. A total of 10 µL of 10-fold diluted extracts or Trolox solution (1500–0 µM) was mixed with 390 µL of distilled water and 3 mL of freshly prepared FRAP reagent. Then, the mixture was vortexed for 10 s and the absorbance was measured at 593 nm after 30 min of incubation in the dark. A duplicate measurement was conducted, and antioxidant power was expressed as µmol of Trolox equivalents per gram of the samples (µmol TE/g).

### 2.12. Color Characteristics

The color attributes were determined using a CR-410 chromameter measuring head (Konica Minolta, Osaka, Japan) based on CIE *L***a***b** color coordinates, *L** (Lightness/darkness; 100 to 0), *a** (positive = redness/negative = greenness), and *b** (positive = yellowness/negative = blueness) at D65 standard illuminant with 2° standard observer angle. The chromameter was calibrated using a white calibration plate. The total color difference (Δ*E*) indicating the magnitude of color difference between control and treated samples was calculated using the following formula:(4)$$∆E={\left[{\left(L*-L_{0}\right)}^{2}+{\left(a*-a_{0}\right)}^{2}+{\left(b*-b_{0}\right)}^{2}\right]}^{\frac{1}{2}}$$
where $L_{0},\;a_{0},$ and $b_{0}$ = blank values of control sample extracted.

The quantitative attribute of color intensity is referred to as Chroma (*C*). Hue (*h*°) is the degree of variation of a particular color in comparison to grey color with the same lightness, such as reddish (0° or 360°), greenish (90°), yellowish (180°), and bluish (270°). a* and b* values can be used to express the chroma and hue angle, respectively, using the equations below [34]:(5)$$\left(C\right)={\left[{\left(a*\right)}^{2}+{\left(b*\right)}^{2}\right]}^{\frac{1}{2}}$$
(6)$$\left(h^{{}^{\circ}}\right)=ArcTan\left(\frac{b*}{a*}\right)$$

Browning index (*BI*) represents a physical indicator of overall changes in browning using the *L**, *a**, and *b** values and calculated using the following equations [35]:(7)$$BI=100\times \left(\frac{X-0.31}{0.17}\right)$$
where
(8)$$X=\frac{\left(a*+1.75L\right)}{\left(5.645L+a*-3.012b*\right)}$$

### 2.13. Statistical Analysis

Data analysis was performed using Minitab V.19 (Minitab Inc., State College, PA, USA). The general linear model (GLM) procedure was utilized to determine the significance of the main dependent variables (drying process and time) and their interactions. One-way analysis of variance (ANOVA) was employed for all statistical analyses. Tukey’s pairwise comparison in ANOVA was used to assess the differences between extracts, with a probability of *p* < 0.05. Additionally, linear Pearson correlations and PCA were utilized to evaluate the correlations between anthocyanins, TPC, TFC, antioxidant activities, and color characteristics. 

## 3. Results

### 3.1. Preliminary Study

A preliminary study was carried out to determine the suitable drying time for each drying process (Table 2). Moisture content < 10% is sufficient for dried MP powder to be microbiologically safe [36], in agreement with Ezzat et al. [37], where depending on the product type, moisture content of powder should be <10% with water activity ranging from 0.1 to 0.4. In this study, the suitable drying times were determined at 36 and 48 h for both freeze-drying and oven-drying as the moisture attained ranged from 3.08 to 8.86%. The initial moisture content of fresh MP was recorded at ~60.39%, which is similar to that in García et al. [38], who obtained ~64.3%.

As explained by Nemzer et al. [39], freeze-drying works by freezing the plant material and then allowing the frozen water inside to sublimate, whereas oven-drying works by exposing it to a continuous flow of hot air where the moisture is transferred out of the plant material by diffusion or a capillary mechanism [40]. On the other hand, sun-drying needed 30 and 40 h of drying to reach moisture content between 8.79 and 7.59%. Sun-drying may be applied under mild drying conditions; however, it decreases the drying rate, which prolongs the drying time. This method is considered a traditional and slow process involving high humidity, haze, weather uncertainties, insects, rodents, and birds, which leads to poor quality of plant materials [20].

### 3.2. LC-MS Analysis

According to the preliminary study, TMAC was detected highest in freeze-dried samples (Table 2). Samples after 36 h of freeze-drying had approximately 3.4-fold higher TMAC compared to control (~0.51 mg/g), while sun-drying at 40 h showed the lowest TMAC (~0.14 mg/g). Although FD48 also recorded better results in TMAC with 1.89 mg/g, FD36 was selected for LC-MS analysis due to lower time and energy consumption during the freeze-drying process. Following that, phenolic compounds including anthocyanins in freeze-dried MP were identified by mass spectrometry. As shown in Table 3, the MS^2^ data show a characteristic fragment ion at *m*/*z* 287.0558, accounting for C3S from the parent ion *m*/*z* 611.16 [M − H]^+^, while the fragment ion at *m*/*z* 287.0537 represents C3G from the parent ion *m*/*z* 449.11 [M − H]^+^.

Notably, *m*/*z* of P3G was not detected, in contradiction to that reported by Zarena and Udaya [15], who found 6.2% of P3G present in MP. Moreover, analysis of anthocyanins by Yenrina et al. [49] also proved the absence of P3G in MP. This is due to the utilization of acidified methanol by Zarena and Udaya [15] in the MP extraction, which facilitates the extraction of P3G. On the other hand, the current study only used 50% aqueous ethanol as a medium for extraction in the water bath shaker. As stated by Albuquerque et al. [50], following different extraction and purification methods has a significant impact on the composition and concentration of phenolic compounds obtained. Thus, they may not be comparable or consistent with the findings by other authors.

Other than anthocyanins, catechin and xanthones including *β*-mangostin, procyanidin dimer, procyanidin trimer, 3-isomangostin, garcimangosxanthone C, gartanin, garcimangosone C, and 9-hydroxycalabaxanthones are among the major phenolic compounds identified in freeze-dried MP. The identification was made according to the parent ion [M − H]^+^ or [M − H]^−^ and fragment ion pattern (MS^2^) found in MP, where the information found in the literature was used for the confirmation of the compounds as presented in Table 3.

### 3.3. Anthocyanins

As shown in Figure 1 and Table 4, significantly (*p* < 0.05) higher C3S (~95%) was quantified using HPLC, followed by C3G (~4.8%), in freeze-dried MP. This is in agreement with the findings by Zarena and Udaya [15], who reported that C3S (76.1%) was the major anthocyanins detected in MP, followed by C3G (13.4%). There were no significant differences in the individual and total anthocyanins between FD36 and FD48. C3S was found 10-fold higher in freeze-dried samples compared to control, with only 0.09–0.10 mg/g of C3G quantified. However, except freeze-dried, C3G content was detected lower than the detection limit in fresh, oven-, and sun-dried MP extracts.

Statistical analysis verified that only drying process and the interaction between drying process and time have a significant (*p* < 0.05) effect on total anthocyanins in dried MP extracts. Based on the HPLC results (Table 4), the highest (*p* < 0.05) total anthocyanins were detected in freeze-dried samples, followed by oven-dried (0.70–0.41 mg/g) and fresh MP (~0.22 mg/g). On the contrary, sun-drying inefficiently preserved the anthocyanins, resulting in 50% degradation compared to fresh MP. This finding is in agreement with Çoklar and Akbulut [51], who found that drying of black grapes using freeze-drying (−110 °C, 48 h) increased 16% of anthocyanins, but reduced 95% of these compounds in both oven- (60 °C, 17 h) and sun-dried (7 days) samples compared to fresh (~1752.94 mg/kg). Drying anthocyanins-rich plants will accelerate the extraction and allow penetration of solvent into cells and extract the targeted compounds while maintaining the chemical properties of anthocyanins [52]. Considering anthocyanins that are sensitive to light and heat, exposing the MP to direct sunlight for longer periods led to the degradation of heat-sensitive compounds, particularly anthocyanins.

The highest total anthocyanins were detected in FD36 (~2.11 mg/g) and FD48 (~2.20 mg/g). This is in agreement with Routray et al. [53], where total anthocyanins in freeze-dried blueberry leaf extract were recorded as the highest compared to microwave-dried and hot air-dried, ranging from 0.26 to 0.79 mg/g. Furthermore, Si et al. [54] reported that freeze-dried raspberry fruits exhibited the highest anthocyanins retention (~0.33 g/kg DW) than hot-air drying, infrared radiation, and microwave-vacuum drying. In contrast, Ling et al. [55] found that freeze-drying (~0.80 mg/g) lowered the value of anthocyanins in seaweeds compared to oven-drying (~1.13 mg/g) at 40 °C. Freeze-drying can minimize the negative effects of high temperatures on thermally sensitive anthocyanins while preserving a higher amount of anthocyanins [56].

In oven-drying, significantly (*p* < 0.05) higher total anthocyanins were recorded after 36 h (~0.70 mg/g) than 48 h (~0.41 mg/g). Since oven-drying involved heating at 45 ± 1 °C, total anthocyanins increased as the MP dried up to 36 h. Gupta et al. [57] found that drying Irish brown seaweed (*Himanthalia elongate*) in a hot-air oven at lower temperatures (25 and 30 °C) resulted in an ongoing reduction in TPC. However, drying at higher temperatures (35 and 40 °C) resulted in an increase in TPC during the first 2 h, after which it began to decline. Additionally, after drying at 40 °C for 2 h, TPC increased by 41%. This increase could be attributed to developmental changes and a wound-like response caused by drying [55].

On the other hand, the lowest total anthocyanins were recorded in sun-dried MP extracts, whereas no significant difference was found between different drying times (30 and 40 h). MP were exposed to intense UV radiation, air, and heat for the extended drying time, leading to loss of color and bioactive compounds. According to Ling et al. [55], when exposed to direct sunlight, anthocyanins compounds migrate out of the plant cell materials, leading to a leaching effect that reduces the yield of anthocyanins in MP. Anthocyanins in MP are stable at temperatures 5 to 50 °C; however, degradation occurs when subjected to higher temperatures [58]. 

### 3.4. Total Phenolic Content (TPC) and Total Flavonoid Content (TFC)

A high level of phenolic compounds mainly found in MP is potentially a rich source of natural antioxidants [59]. As shown in Table 5, TPC in dried MP ranged between 43.16 to 94.05 mg GAE/g after undergoing different drying processes and times. The significantly (*p* < 0.05) highest TPC (~94.05 mg GAE/g) was observed in FD36, whereas the lowest content was attained after 48 h of oven-drying (~43.16 mg GAE/g). In comparison to fresh MP, freeze-drying increased total phenolics by more than 60% after drying. It is known that phenolic compounds are susceptible to heat treatment, and the application of a very low temperature during freeze-drying was considered the most efficient method for preserving phenolic compounds [60]. Furthermore, high levels of phenolic compounds were extracted due to the rupture of the cell structure that formed ice crystals in the tissue matrix during freeze-drying [61]. 

According to Azman et al. [19], the degradation of phenolic compounds in blackcurrant (*Ribes nigrum* L.) pomace was observed when subjected to hot-air oven drying at high temperatures (50 and 130 °C). This was attributed to the extended processing time and high drying temperature. In addition, a significant decrease was found in TPC in hot-air drying of persimmon at 55 °C, as reported by Kayacan et al. [62] with TPC results of freeze-drying (262.4 mg GAE/100 g) close to fresh persimmon (265.1 mg GAE/100 g). Mediani et al. [63] reported on the TPC of *Cosmos caudatus* under different drying processes where freeze-drying (25.3 mg GAE/100 g) showed a better result compared to oven-drying (19.44 mg GAE/100 g) as it caused the heat treatment to degrade bioactive compounds in plants. According to Çoklar and Akbulut [51], despite TPC of grapes oven-dried at 60 °C being lower than that of freeze-dried and fresh grapes, the oven-drying may have protected the phenolic content from prolonged oxidative degradation compared to sun-drying, which is similar to the findings of this study.

The main groups of polyphenolic compounds in MP are known as anthocyanins, isoflavones, catechins, lignans, phenolic acids, and flavonoids [64]. Similar to TPC, most of the drying processes resulted in significantly higher TFC (196–621 mg CE/g), except for SD30 (Table 5). FD36 increased by 72% after drying, with a 9.7% loss in SD30 compared to fresh MP (~176 mg CE/g). Even though oven-drying was processed at a higher temperature, the value of oven-drying showed better TFC retention compared to sun-drying. Gao et al. [65] explained that the losses of phenolic compounds were caused by slow deactivation of degradative enzymes that degraded before the plant materials completely dried during the process. The prolonged exposure (30 to 40 h) to oxygen and light under direct sun at a slower drying rate also reduces the phytochemical content in MP. 

Moreover, the convection heating of oven-drying could result in the breakdown of cells in MP, causing the increase of flavonoids in the extract. A study by Miletić et al. [66] on plum cultivars “Valjecka” and “Mildora” by air-drying at 90 °C increased TFC by 5%. Moreover, Annegowda et al. [67] reported a decrease in TFC after oven-drying (24 h, 60 °C) in ripe papaya by 67% but an increase of up to 105% in raw papaya. In contrast, TFC in ripe papaya increased by two-fold after 24 h freeze-drying, while a 50% decrease was detected in raw papayas. In this study, strong positive correlations (*p* < 0.05) between TPC (r = 0.880) and TFC (r = 0.951) with total anthocyanins were obtained, indicating that anthocyanins are among the compounds that contributed to high TPC and TFC. Following this study, freeze-drying showed better retention of flavonoids, thus proving the disadvantages of high temperature and prolonged time for dried MP. As explained by Capanoglu [68], breakdown of cell walls may result in better extractability of compounds from the plant, which explained the increase in TFC after drying.

### 3.5. Antioxidant Activities

MP are abundant in bioactive compounds with good antioxidant activities, such as anthocyanins and xanthones [69]. The antioxidant activity in the dried MP extract was effectively evaluated using the TEAC and FRAP assays. The statistical analysis confirmed that the drying process, time, and their interaction had a significant (*p* < 0.05) impact on the antioxidant capacity and reducing power of the dried MP. This is in accordance with the findings by Ibrahim et al. [70] where antioxidant activity and total phenolic content of MP were shown to be affected by drying time and temperature.

As presented in Table 5, the antioxidant capacity in dried MP extracts ranges between 201.40 to 802.40 μmol TE/g using TEAC assay. Freeze-drying showed significantly (*p* < 0.05) higher antioxidant capacity, followed by oven-drying, while the lowest activities were recorded in sun-dried and fresh MP. Among freeze-dried MP, higher activity (*p* < 0.05) was recorded in FD48 (~802.40 μmol TE/g) than FD36 (~785.90 μmol TE/g). In addition, significantly higher reducing power was detected in FD36 (~1154.50 μmol TE/g) and FD48 (~1017.50 μmol TE/g) than in other drying methods (Table 5). 

Sze Lim et al. [71] studied different parts of mangosteen (pericarp, pulp, and seed) extracts, and the results for antioxidant capacity (122.00 μmol TE/g) and reducing power (~18.99 mM ferrous sulphate equivalent/g DW) in pericarps were almost similar to this study. However, variations of mangosteen, extraction methods, and solvent used also play an important part in the quantification of antioxidant activities of MP. 

Sogi et al. [61] studied mango cubes under four different drying treatments including freeze-drying, hot-air drying, vacuum-drying, and infra-red drying. The findings concluded that freeze-dried mango cubes exhibited significantly higher (*p* < 0.05) antioxidant capacity (88.6 μmol TE/g) compared to drying methods that involved heat, which caused a decrease in antioxidant properties. Routray et al. [53] obtained the highest activity from freeze-dried blueberry leaves, followed by microwave-assisted hot-air dried and hot-air dried, at three different temperatures (45, 60 and 75 °C). Oven-dried Chinese cabbage at 100 °C for 48 h reduced the reducing power to 1.38 mg TEAC/100 g compared to solar drying, microwave drying, and sun-drying with freeze-dried samples showing the highest reducing power (~4.49 mg TEAC/100 g) [72]. 

The other non-phenolic antioxidants, such as ascorbic acid, that have larger redox potential can be retained by the non-thermal freeze-drying process, which contributes to antioxidant activities [73]. The antioxidant activities of this study showed similar results to Feng et al. [74], where freeze-drying of apple samples after drying exhibited the highest antioxidant activities compared to microwave-vacuum drying (70 °C) and freeze-drying-explosion puffing drying (70 °C). Previous studies also reported the effects of prolonged heat treatment and heat intensity resulting in the degradation of bioactive components and losses of antioxidant activities [75]. However, the variation in mangosteen used, extraction methods, and solvent used plays an important part in the quantification of antioxidant activities of MP.

Low-temperature processing during freeze-drying can preserve the flavonoids that are responsible for antioxidant activity by inhibiting their biological activity [76]. Therefore, exposure to high temperatures for a longer time especially during oven-drying (48 h) and sun-drying for 30 and 40 h significantly affected the antioxidant activities. A strong correlation (r = 0.976, *p* < 0.05) was observed between TEAC and reducing power, proving an accurate measurement of antioxidant activity. Furthermore, strong correlations (*p* < 0.05) were recorded between TEAC and reducing power with total anthocyanins (r = 0.941, r = 0.945), TPC (r = 0.935, r = 0.918), and TFC (r = 0.970, r = 0.978), suggesting that other than anthocyanins, TPC and TFC are also responsible for high antioxidant activity in MP. 

### 3.6. Color Analysis 

Color can be a visual indicator for the drying process. Comparison of color characteristics of dried MP extracts and control (fresh MP) were expressed in terms of color values (*L**, *a**, *b**), total color difference (*ΔE*), chroma (*C*), hue angle (*h*°), and browning index (*BI*), as shown in Table 6 and Figure 2, respectively. Other than anthocyanins, TPC, TFC, and antioxidant activities, color characteristics of dried MP extract were also significantly (*p* < 0.05) affected by drying process, time, and their interactions. 

In this study, all drying processes resulted in better color retention than fresh MP. Dried MP, particularly freeze-dried samples, had a more concentrated, reddish-purpled color, indicating darkness, redness, and less yellowness of the extract. Subsequently, these *L**, *a**, and *b** values also contributed to significantly (*p* < 0.05) lower chroma and hue angle values, but higher total color difference. The hue angles of FD36, FD48, and OD36 were the lowest (0.25°–0.29°), suggesting the redness of dried MP, which ranged between 0 and 30° [77].

The decrease in chroma values was noted for freeze-dried samples, as shown by the lower *a** and *b** values. Chroma values increased with an increase in the processing temperature. These findings differ from the study by Michalska et al. [78], where drying of blackcurrant pomace decreased in *L** and increased in *a** and *b** values after freeze-drying. As stated in their study, freeze-drying accelerates the release of reddish pigments from the pomace when lower pressure is applied compared to microwave-drying and convective drying. Higher temperatures and longer times during oven-drying and sun-drying resulted in the degradation of anthocyanins. The color of anthocyanins changed from red to orange after heating up to 40 °C even at pH 3 and pH 4, according to research by West and Mauer [79].

The highest *L** and *b** values after sun-drying indicated higher lightness and yellowness of MP extract. The purity of color is indicated by chroma or intensity. As depicted in Figure 2, the fresh MP (control) and samples subjected to oven-drying for 48 h and sun-drying had higher chroma, resulting in a brighter color. Due to higher anthocyanins concentration, freeze-dried extracts particularly at 36 h had lower purity and darker color, making them suitable to be used as natural food colorants. 

Negative correlations established between total anthocyanins and *L** (r = −0.800) and *b** (r = −0.823) proved that *L** and *b** values decreased when total anthocyanins increased. To preserve the color and quality of anthocyanins in dried MP after drying, it is important to minimize browning, which reduces their stability and bioavailability. Interestingly, different drying processes resulted in a significantly (*p* < 0.05) higher browning index compared to the fresh MP, considering the higher *b** values contributed to more yellowness related to browning reactions. In this study, freeze-drying exhibited the lowest browning index, similar to the study by Feng et al. [74]. They found that freeze-dried apples had significantly (*p* < 0.05) lower browning index compared to microwave-vacuum drying (70 °C) and freeze-drying-explosion puffing drying (70 °C). Overall, only the interaction between the drying process and time appeared to influence (*p* < 0.05) the browning index in dried MP extracts.

### 3.7. Principal Component Analysis (PCA)

PCA was performed to evaluate the relationships between drying process and time on the total anthocyanins, TPC, TFC, TEAC, reducing power, as well as color characteristics (*L**, *a**, and *b**) in dried MP (Figure 3a). The distribution of quality attributes was defined by 87.4% PC1 and 8.6% PC2 (Figure 3b). In addition, Figure 3c shows that the far right of the plot, FD36, FD48, and OD36, corresponds to the highest value of total anthocyanins, TPC, TFC, TEAC, and reducing power. Contrary to that, oven-dried and sun-dried samples were placed on the opposite side, favoring the *a** values, showing increased yellowness in dried MP related to the browning effect after drying due to prolonged time and high temperature. This result suggests that FD36, FD48, and OD36 have similar effects on the principal components, but significant negative effects were displayed by OD48 and sun-dried MP samples. Overall, freeze-drying is considered the most efficient method for preserving the phenolic compounds and antioxidant activities in MP.

## 4. Conclusions

Factors such as drying temperature, time, and their interactions significantly affected the phenolics compounds, antioxidant activities, and color of MP extracts. Exposure of MP under extended drying time and heat is not appropriate for the temperature- and oxygen-sensitive compounds, particularly anthocyanins. Therefore, in comparison to oven-drying and sun-drying, freeze-drying for 36 h efficiently preserved the anthocyanins, TPC, TFC, and antioxidant activities in MP due to low-temperature drying and less energy and time consumption. As confirmed through LC-MS analysis, only C3S and C3G but no P3G were detected in all drying processes due to different extraction methods used. According to HPLC analysis, C3S (~95%) was detected higher than C3G (~4.8%) in freeze-dried MP. Yet, C3G was lower than the detection limit in fresh, oven-dried, and sun-dried MP extracts. Correlation results revealed that anthocyanins were among the highest contributors to TPC, TFC, antioxidant activities, as well as color characteristics. Lower *L** and chroma values in freeze-dried MP indicated a darker extract and thus its potential to be utilized as a natural colorant. Overall, these findings could be valuable to the colorants or ingredients industry for incorporation into food, nutraceutical, and pharmaceutical products.

## Figures and Tables

**Figure 1 foods-12-02351-f001:**
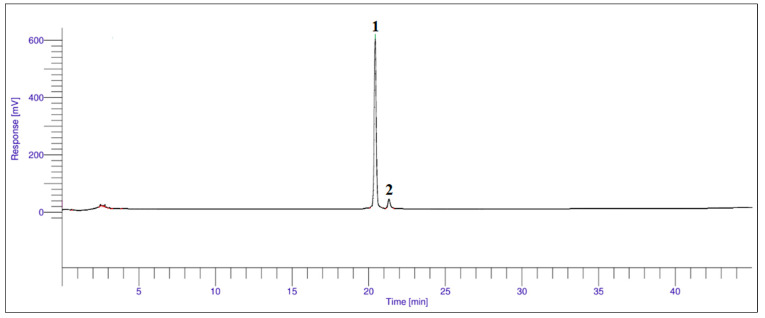
Typical HPLC chromatograms showing anthocyanins in dried MP extracts detected at 520 nm. (1) cyanidin-3-*O*-sophoroside and (2) cyanidin-3-*O*-glucoside.

**Figure 2 foods-12-02351-f002:**
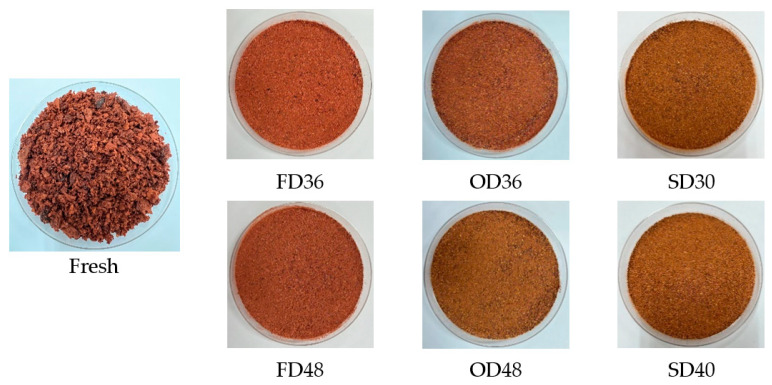
The physical appearance of fresh and dried MP powder after different drying processes and times. FD: freeze-dried, OD: oven-dried, SD: sun-dried.

**Figure 3 foods-12-02351-f003:**
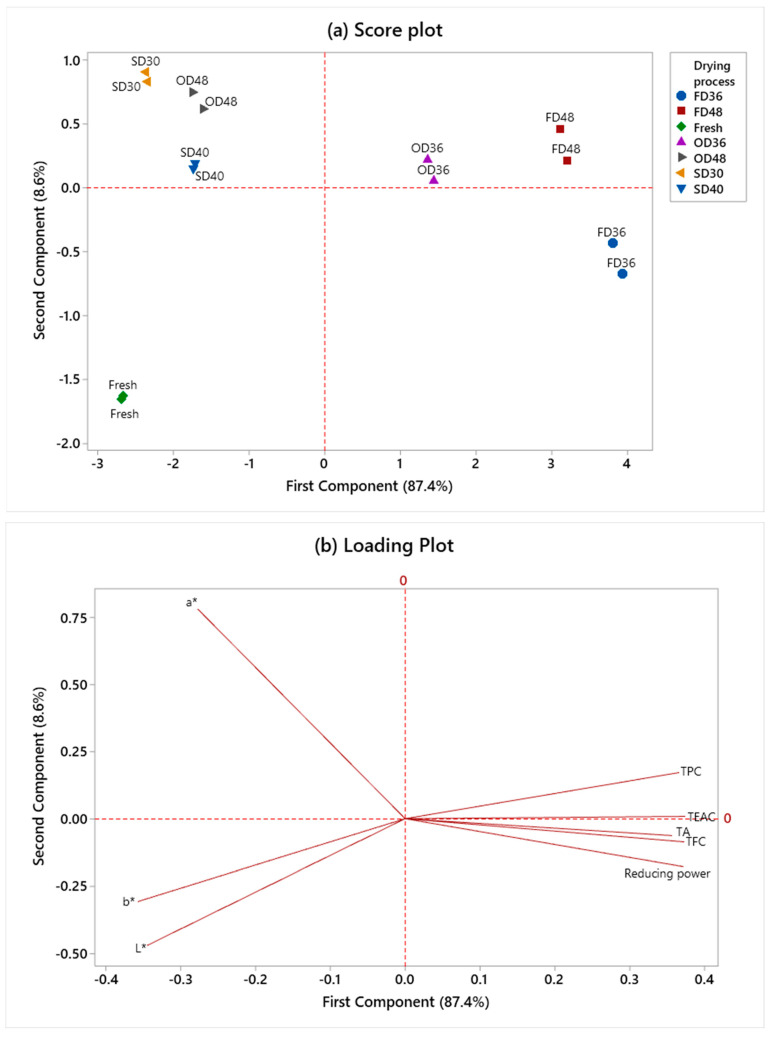
(**a**) Score plot, (**b**) loading plot, and (**c**) bi-plot of principal component analysis based on the effect of different drying processes and times on dried MP extract. FD: freeze-dried; OD: oven-dried; SD: sun-dried; TA: total anthocyanins; TPC: total phenolic content; TFC: total flavonoid content; TEAC: Trolox equivalent antioxidant capacity. *L**: lightness/darkness; 100 to 0; *a**: positive = redness/negative = greenness; *b**: positive = yellowness/negative = blueness. Results are expressed as mean ± standard deviation (*n* = 2).

**Table 1 foods-12-02351-t001:** Determination coefficient (R^2^), limits of detection (LOD), and limit of quantification (LOQ) of individual anthocyanins as quantified by HPLC.

Anthocyanins	R^2^	LOD (mg/mL)	LOQ (mg/mL)
C3S	0.9985	0.01	0.03
C3G	0.9993	0.01	0.02
P3G	0.9997	0.01	0.02

C3S: cyanidin-3-*O*-sophoroside; C3G: cyanidin-3-*O*-glucoside; P3G: pelargonin-3-*O*-glucoside.

**Table 2 foods-12-02351-t002:** Effect of different drying processes and times on moisture content, water activity, and total monomeric anthocyanins content in MP extract.

Drying Process and Time (h)	Moisture Content (%)	Water Activity (A_w_)	Total Monomeric Anthocyanins Content (mg/g)
Fresh (Control)	60.39 ± 0.38 ^a^	0.97 ± 0.00 ^a^	0.51 ± 0.01 ^d^
FD36	7.48 ± 0.73 ^b^	0.59 ± 0.01 ^b^	2.28 ± 0.01 ^a^
FD48	3.08 ± 0.11 ^d^	0.35 ± 0.01 ^f^	1.89 ± 0.02 ^b^
OD36	8.86 ± 0.08 ^b^	0.55 ± 0.00 ^c^	0.97 ± 0.06 ^c^
OD48	5.17 ± 0.81 ^c^	0.41 ± 0.01 ^e^	0.44 ± 0.04 ^d^
SD30	8.79 ± 0.26 ^b^	0.52 ± 0.00 ^d^	0.23 ± 0.05 ^e^
SD40	7.59 ± 0.28 ^b^	0.56 ± 0.00 ^c^	0.14 ± 0.01 ^e^

FD: freeze-dried; OD: oven-dried; SD: sun-dried. Results are expressed as mean ± standard deviation (*n* = 2). Values with the same letter ^a–f^ in each row are not significantly different (*p* < 0.05).

**Table 3 foods-12-02351-t003:** Metabolites identified in mangosteen pericarp ethanolic extracts using LC-MS/MS in positive and negative ionization modes.

No.	Rt (min)	Compound Name	Chemical Structure	MS^+^ (*m*/*z*)	MS^−^ (*m*/*z*)	MS^2^ (*m*/*z*)	References
1.	9.34	*β*-mangostin	C_25_H_26_O_6_		423.18	109.0283, 151.0026, 261.0404, 423.0945	[41]
2.	15.86	Procyanidin dimer	C_30_H_26_O_12_		577.14	125.0228, 161.0231, 245.0820, 289.0707, 407.0749, 425.0887	[41,42,43]
3.	16.43	Cyanidin-3-*O*-sophoroside	C_27_H_31_O_16_	611.16		287.0558	[41]
4.	18.10	Procyanidin trimer	C_45_H_36_O_18_		865.20	125.0230, 161.0233, 289.0714, 407.0761	[41]
5.	18.20	Cyanidin-3-*O*-glucoside	C_21_H_21_O_11_	449.11		287.0537	[41,42]
6.	20.65	Catechin	C_15_H_14_O_6_		289.07	109.0283, 125.0233, 151.0386, 203.0711, 245.0815, 289.0724	[42,43,44,45]
7.	33.34	Quercetin-3-*O*-glycoside	C_21_H_20_O_12_		463.21	151.0021, 178.9979, 271.0231, 300.0262	[42,43,44,46]
8.	50.95	3-isomangostin	C_24_H_24_O_6_		427.18	351.0872, 369.0969, 409.1279, 427.1374	[47]
9.	52.82	Garcimangosxanthone C	C_19_H_22_O_6_		345.13	244.0367, 275.0919, 299.0910, 344.0858	[41]
10.	52.96	Gartanin	C_23_H_24_O_6_		395.15	271.0252, 283.0235, 297.0396, 339.0872, 395.1487	[47,48]
11.	53.55	Garcimangosone C	C_23_H_24_O_7_	411.18		271.0592, 299.0550, 337.1065, 355.1176, 411.1775	[48]
12.	53.71	9-hydroxycalabaxanthone	C_24_H_24_O_6_		407.15	339.0883, 351.0880, 377.1392, 394.1425, 409.1641,	[47,48]

**Table 4 foods-12-02351-t004:** Total anthocyanins content in dried MP under different drying processes and times as quantified by HPLC.

Drying Process and Time (h)	Anthocyanins (mg/g)			
	C3S	C3G	P3G	Total
Fresh (Control)	0.21 ± 0.03 ^cd^	<LOD	ND	0.22 ± 0.04 ^d^
FD36	2.02 ± 0.02 ^a^	0.09 ± 0.01 ^a^	ND	2.11 ± 0.02 ^a^
FD48	2.09 ± 0.12 ^a^	0.10 ± 0.00 ^a^	ND	2.20 ± 0.12 ^a^
OD36	0.68 ± 0.01 ^b^	<LOD	ND	0.70 ± 0.01 ^b^
OD48	0.39 ± 0.00 ^c^	<LOD	ND	0.41 ± 0.00 ^c^
SD30	0.14 ± 0.00 ^d^	<LOD	ND	0.14 ± 0.00 ^d^
SD40	0.15 ± 0.01 ^d^	<LOD	ND	0.15 ± 0.01 ^d^

FD: freeze-dried; OD: oven-dried; SD: sun-dried; C3S: cyanidin-3-*O*-sophoroside; C3G: cyanidin-3-*O*-glucoside; P3G: pelargonidin-3-*O*-glucoside. Results are expressed as mean ± standard deviation (*n* = 2). Values with the same letter ^a–d^ in each row are not significantly different (*p* < 0.05).

**Table 5 foods-12-02351-t005:** Effect of different drying processes and times on total phenolic content, total flavonoid content, and antioxidant activities in MP extract.

Drying Process and Time (h)	Total Phenolic Content (mg GAE/g)	Total Flavonoid Content (mg CE/g)	Trolox Equivalent Antioxidant Capacity (µmol TE/g)	Reducing Power (µmol TE/g)
Fresh (Control)	31.32 ± 0.52 ^g^	176.00 ± 0.88 ^f^	234.40 ± 1.41 ^e^	495.00 ± 8.49 ^d^
FD36	94.05 ± 0.45 ^a^	621.00 ± 2.65 ^a^	785.90 ± 7.78 ^a^	1154.50 ± 4.95 ^a^
FD48	86.39 ± 0.52 ^b^	568.50 ± 0.88 ^b^	802.40 ± 4.24 ^a^	1017.50 ± 10.61 ^b^
OD36	73.76 ± 1.35 ^c^	386.63 ± 7.07 ^c^	607.90 ± 24.75 ^b^	687.50 ± 34.65 ^c^
OD48	43.16 ± 1.12 ^f^	196.63 ± 1.77 ^e^	365.90 ± 2.12 ^c^	521.00 ± 8.49 ^d^
SD30	48.86 ± 1.12 ^e^	159.13 ± 1.77 ^g^	201.40 ± 1.41 ^e^	411.00 ± 7.78 ^e^
SD40	55.98 ± 0.45 ^d^	271.00 ± 4.42 ^d^	306.40 ± 0.00 ^d^	528.00 ± 50.21 ^d^

FD: freeze-dried, OD: oven-dried, SD: sun-dried, GAE: gallic acid equivalent; CE: catechin equivalent; TE: Trolox equivalent. Results are expressed as mean ± standard deviation (*n* = 2). Values with the same letter ^a–g^ in each row are not significantly different (*p* < 0.05).

**Table 6 foods-12-02351-t006:** Color properties of MP extracts for different drying processes and times.

Drying Process and Time (h)	Color						Browning Index
	*L**	*a**	*b**	*C*	*h*°	Δ*E*	
Fresh (Control)	44.47 ± 0.00 ^a^	22.48 ± 0.04 ^b^	29.21 ± 0.01 ^a^	36.85 ± 0.03 ^a^	0.91 ± 0.00 ^a^	-	40.27 ± 0.05 ^d^
FD36	26.36 ± 0.29 ^e^	19.65 ± 0.71 ^c^	5.04 ± 0.24 ^f^	20.28 ± 0.75 ^d^	0.25 ± 0.00 ^f^	465.83 ± 13.17 ^a^	49.43 ± 1.14 ^c^
FD48	26.46 ± 0.44 ^e^	22.88 ± 0.78 ^b^	6.12 ± 0.29 ^e^	23.68 ± 0.81 ^c^	0.26 ± 0.00 ^f^	434.62 ± 14.41 ^a^	56.46 ± 0.90 ^a^
OD36	28.29 ± 0.25 ^d^	21.92 ± 0.47 ^b^	6.59 ± 0.03 ^e^	22.88 ± 0.45 ^c^	0.29 ± 0.01 ^e^	391.94 ± 4.45 ^b^	51.55 ± 1.32 ^bc^
OD48	35.43 ± 0.38 ^c^	27.05 ± 0.54 ^a^	18.68 ± 0.42 ^d^	32.87 ± 0.68 ^b^	0.60 ± 0.00 ^d^	109.71 ± 5.51 ^c^	54.22 ± 0.46 ^ab^
SD30	35.61 ± 0.01 ^c^	27.40 ± 0.16 ^a^	20.93 ± 0.09 ^c^	34.48 ± 0.18 ^b^	0.65 ± 0.00 ^c^	88.43 ± 0.06 ^c^	55.26 ± 0.27 ^a^
SD40	37.38 ± 0.20 ^b^	25.88 ± 0.26 ^a^	23.21 ± 0.18 ^b^	34.76 ± 0.31 ^ab^	0.73 ± 0.00 ^b^	51.13 ± 1.64 ^d^	51.23 ± 0.21 ^bc^

*L**: (Lightness/darkness; 100 to 0); *a**: (positive = redness/negative = greenness); *b**: (positive = yellowness/negative = blueness); *C*: chroma; *h◦*: hue angle; Δ*E*: total color difference. Figures in parentheses indicate the standard deviation (*n* = 2). Values with the same letter ^a–f^ in each row are not significantly different (*p* < 0.05).

## Data Availability

Data are contained within the article.

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
