# Peer review of "Influence of Different Drying Methods on Anthocyanins Composition and Antioxidant Activities of Mangosteen (Garcinia mangostana L.) Pericarps and LC-MS Analysis of the Active Extract"

_foods, 2023, doi:10.3390/foods12122351_

Round 1
Reviewer 1 Report
The paper is written in an interesting and exhaustive way. I haven't noticed any major inaccuracies. Perhaps the correctness of the English language should be checked more carefully. I'm not a native speaker, so I can't say for sure.
Author Response
Dear Reviewer 1,
Please see the attachment.
Thank you.

Reviewer 2 Report
The paper titled “Comparison Effects of Sun, Oven, and Freeze-Drying Methods on Anthocyanins Composition and Antioxidant Activities of Mangosteen (Garcinia mangostana L.) Pericarps”, represent a well written and detailed study on the application of different drying processes and times on the antioxidant compounds of Mangosteen Pericarps. I provide only a few indications to improve the work:
1) revise the title if you think it appropriate.
2) review the English form throughout the text, particularly in the abstract section.
3) Pag. 3, lines 105-106: Specify what is indicated with "at maturity index 6". Enter more information in the text.
4) Pag. 3, line 133: 180 rpm, it would seem a low value, did you write it wrong?
5) Results: pag. 7. Enter more discussion about the Total Monomeric Anthocyanins Content. Add considerations about the highest values in FD36 and FD48 samples.

I think the article requires moderate editing of English language.
Author Response
Dear Reviewer 2,
Please see the attachment.
Thank you.

Reviewer 3 Report
The manuscript "Comparison Effects of Sun, Oven, and Freeze-Drying Methods on Anthocyanins Composition and Antioxidant Activities of Mangosteen (Garcinia mangostana L.) Pericarps" submitted at Foods journal is interesting. Factors such as drying temperature, time, and the effect of their interaction on phenolics compounds, antioxidant activities, and color were assessed. Waste utilization is a hot topic nowadays.
Moreover, Garcinia mangostana L. is an endemic evergreen tree species grown in tropical rainforest countries, such as Malaysia, Thailand, and Indonesia, and based on the abovementioned it is worth to be explored.
In this regard, the manuscript is valuable.
Some minor comments should be taken into account in order to improve the text:
- The fresh fruits were selected for uniformity of color at maturity index 6. – Reference?
- Lines 209-217 – indicate how the results are expressed
- Why are TPC and TFC expressed in grams and AOA is estimated by FRAP analysis based on 100 grams? On the other hand, it is strange that only two methods were applied to evaluate AOA and DPPH Radical Scavenging Activity is expressed as % inhibition, which does not allow comparison of results.
- The visualization of Table 6 and the one presented in Figure 2 are duplicated and the figure is more informative.
Author Response
Dear Reviewer 3,
Please see the attachment.
Thank you.
